# Microbial Populations in Ruminal Liquid Samples from Young Beefmaster Bulls at Both Extremes of RFI Values

**DOI:** 10.3390/microorganisms11030663

**Published:** 2023-03-06

**Authors:** Nelson Manzanares-Miranda, Jose F. Garcia-Mazcorro, Eduardo B. Pérez-Medina, Anakaren Vaquera-Vázquez, Alejandro Martínez-Ruiz, Yareellys Ramos-Zayas, Jorge R. Kawas

**Affiliations:** 1Centro de Investigación en Producción Agropecuaria (CIPA), Universidad Autónoma de Nuevo León (UANL), Linares 67700, Mexico; 2Research and Development, MNA de Mexico, San Nicolas de los Garza 66477, Mexico; 3Laboratorio de Biología Molecular, Laboratorio Central Regional del Norte (LCRN), Guadalupe 67100, Mexico; 4Posgrado Conjunto Agronomía-Veterinaria, UANL, General Escobedo 66054, Mexico; 5Facultad de Agronomía, UANL, General Escobedo 66054, Mexico

**Keywords:** beef cattle, feed efficiency, residual feed intake, ruminal microbiota, *Succiniclasticum*

## Abstract

The gut microbiota is involved in the productivity of beef cattle, but the impact of different analysis strategies on microbial composition is unclear. Ruminal samples were obtained from Beefmaster calves (*n* = 10) at both extremes of residual feed intake (RFI) values (5 with the lowest and 5 with the highest RFI) from two consecutive days. Samples were processed using two different DNA extraction methods. The V3 and V4 regions of the 16S rRNA gene were amplified using PCR and sequenced with a MiSeq instrument (Illumina). We analyzed 1.6 million 16S sequences from all 40 samples (10 calves, 2 time points, and 2 extraction methods). The abundance of most microbes was significantly different between DNA extraction methods but not between high-efficiency (LRFI) and low-efficiency (HRFI) animals. Exceptions include the genus *Succiniclasticum* (lower in LRFI, *p* = 0.0011), and others. Diversity measures and functional predictions were also mostly affected by DNA extraction methods, but some pathways showed significant differences between RFI levels (e.g., methylglyoxal degradation, higher in LRFI, *p* = 0.006). The results suggest that the abundance of some ruminal microbes is associated with feed efficiency and serves as a cautionary tale for the interpretation of results obtained with a single DNA extraction method.

## 1. Introduction

In animal production systems, feed efficiency refers to the ability of an animal to turn feed ingredients into products for human consumption, such as meat and milk. Feed efficiency is a complex multifaceted trait, under the control of several interrelated biological processes and management regimens [1,2]. These biological processes include those associated with the animal (e.g., energy metabolism) and also the composition and function of the digestive microbiota [1]. While the relationship between the gut microbiota and the health and productivity of animals has been well studied, more details are still needed to better understand the nature and characteristics of this relationship.

There are key characteristics of the gut microbiota that make this subject difficult to investigate in the context of feed efficiency, such as its relationship with the animal’s genetics. One study of beef cattle (*n* = 48) from three breeds showed that the differential microbial features observed between efficient and inefficient steers tended to be specific to breeds, suggesting that interactions between host genotype and the rumen microbiome contribute to the variations in feed efficiency [3]. Another, larger study of 709 beef cattle also showed that breed, sex, and diet were major factors associated with the variation in rumen microbiota [4]. Moreover, it has been shown that high-RFI (residual feed intake) animals spent about 10 min longer eating than their more efficient, low-RFI contemporaries [2], and that growing beef heifers with low RFI had a lower occurrence of non-feeding events (i.e., where cattle are at the feed face but do not consume any feed, [5]), thus suggesting additional factors (e.g., anatomical or physiological) related to feed efficiency that are not directly linked with either genetics or the gut microbes. Finally, the gut microbiota is relatively constant over time, but it may also show patterns of variation that could potentially mask dietary and even host genetics effects [6,7].

The gut microbiota is affected by many factors, but the data we obtain from gut microbial populations in ruminants and other animals is also dependent on methodological factors, such as the source of material for DNA extraction [8], preservation methods of samples [9], DNA extraction methods [10,11], PCR primer choice [12], the data analysis strategies employed [13], and statistical considerations of the resulting data [14]. The variations among methodologies are important because they can provide different views of the structure and composition of microbial populations. The objective of this study was to investigate microbial populations in ruminal contents in beef cattle with LRFI (more efficient) and HRFI (less efficient) using two different DNA extraction methods.

## 2. Materials and Methods

### 2.1. Experimental Animals

This study was conducted in compliance with the current Mexican legislation (NOM-062-ZOO-1999) and revised by the Committee of Animal Research and Experimentation (CARE) at MNA de México (Protocol # 04132021). A total of 33 young beefmaster bulls and 18 calves were included in a feed efficiency test at the Centro de Investigación en Producción Agropecuaria (CIPA) of the Universidad Autonoma de Nuevo Leon (UANL). The adaptation period lasted 14 days and the evaluation period was 78 days. Animals were individually fed a diet that as dry matter consisted of 50% ground corn, 16% Klein grass hay, 10% wet distillers’ grains, 8% cane molasses, 2.5% vitamin and mineral mix, and other ingredients (Appendix A). On day 5 after the end of the evaluation period, we obtained the RFI values from a GrowSafe system for tracking the feed consumption of individual animals [15] and ruminal liquid samples were obtained from 10 calves at both extremes of RFI values (5 with the lowest and 5 with the highest RFI) from two consecutive days (Day 1 and Day 2, to consider inter-day variation) using SELEKT equipment (Nimrod Veterinary Products Ltd., Moreton-in-Marsh, Gloucestershire, UK). The steel collector tip in this equipment does not allow the passage of solids, which can clog the hose, and samples were not filtered because this may discard some microbial populations. Ruminal samples (~30 mL) were placed in a 50 mL tube containing 10 mL of ethanol at 95% for better microbiome stability [9], and frozen at −20 °C until DNA extraction and volatile fatty acid (VFA) analysis.

### 2.2. DNA Extraction

DNA extraction was performed at the Laboratorio Central Regional del Norte using bead beading coupled with two available commercial kits: PureLink Genomic DNA kit (Invitrogen, Thermo Fisher Scientific, Waltham, MA, USA, method A) and Wizard Genomic DNA Purification kit (PROMEGA, Madison, WI, USA, method B). Briefly, samples were thawed at room temperature and four aliquots of 1.5 mL (2 for each method) were obtained from each sample (*n* = 20, 10 calves with two time points). The samples were centrifuged at 13,000 RPM for 5 min and the supernatant discarded. The pellets in each set of 2 tubes were mixed in one tube which was the source material for each DNA extraction. Approximately 100 µL of silica beads were added to the tubes, mixed in a FastPrep^®^-24 (Santa Ana, CA, USA) equipment, and the subsequent steps were performed using the instructions included in the user’s manuals. Both methods use purification columns and are similar in their procedures with the exception of a longer incubation time with proteinase K in method A. DNA was quantified using a NanoDrop and visualized in an agarose gel for a qualitative assessment of DNA quality. DNA samples were shipped to the National Laboratory of Genomics for Biodiversity (LANGEBIO, CINVESTAV, Irapuato, Mexico) for further PCR and 16S rRNA gene sequencing.

### 2.3. PCR and Sequencing

PCR was performed using primers 341F (5′-CCTACGGGNGGCWGCAG-3′) and 785R (5′-GGACTACHVGGGTATCTAATCC-3′), covering the semi-conserved regions V3 and V4 of the 16S rRNA gene (approximately 463 nucleotides, from nucleotide 341 to nucleotide 804). The PCR products were sequenced using a MiSeq instrument (Illumina, San Diego, CA, USA) at LANGEBIO.

### 2.4. Bioinformatics

The results were analyzed using QIIME2 v.2021.11 [16]. Quality filtering was performed using DADA2 [17] using 120 nucleotides to remove low-quality regions of the sequences. The method to remove chimeras in the DADA2 plugin was consensus (chimeras are detected in samples individually, and sequences found chimeric in a sufficient fraction of samples are removed). The output feature table was filtered to remove features appearing in less than 4 samples and with less than 20 in frequency [18]. The filtered table was used for taxonomic assignments, and we did not remove any taxa (e.g., Cyanobacteria and Chloroflexi, note that the relevance of these and other taxa in gut microbial ecology is debatable) unless there were issues with low prevalence. The filtered table was also used for alpha and beta diversity analyses using several metrics in the diversity plugin of QIIME2.

### 2.5. Microbial Taxa Abundance

The relative proportions of 16S reads have historically been the data of choice to perform comparisons of microbial taxa in studies of gut microbiota in ruminants [19,20,21] and other animal species [13]. However, it is well known that relative abundance can lead to spurious correlations, originally pointed out by Pearson more than a century ago [22]. We performed a series of experiments to test the performance of the centered log-ratio (clr) transformation [23] and applied this transformation to the raw number of sequences obtained from the filtered table (see “Centered log-ratio transformation” in Appendix A). Transformations were performed at each phylogenetic level separately.

### 2.6. Prediction of Functional Profiles

We used PICRUSt2 [24] for the prediction of metagenome functions based on 16S marker gene sequencing profiles. The filtered feature table was used for this analysis. The resulting pathways counts were clr-transformed prior to the statistical comparison between LRFI and HRFI (*n* = 20 each).

### 2.7. Analysis of Volatile Fatty Acids

Acetic, propionic, and butyric fatty acids were measured using a standard methodology outlined by M.L. Galyean from Texas Tech University in his manual on Laboratory Procedures in Animal Nutrition Research, in a flame ionization detector in a Varian 3400 CX gas chromatograph (Palo Alto, CA, USA). Briefly, ruminal fluid samples were centrifuged, and 5 mL of the supernatant was mixed with 1 mL of meta-phosphoric acid-2EB solution. The mixture was kept in cold, centrifuged, and the supernatant used for GLC injection.

### 2.8. Statistical Analysis

Productive parameters were compared using a Mann–Whitney test. Taxa abundance based on clr-transformed data was analyzed with the MIXED procedure (PROC MIXED) in SAS University Edition (release 3.81) using the clr-transformed data from each taxon as the dependent variable, and day of sampling, DNA extraction method, and RFI, as independent variables (i.e., fixed effects), without random effects. In the case of having residuals with non-normal distributions, the non-parametric 1-way procedure (PROC NPAR1WAY) was used with the disadvantage of analyzing independent variables separately. Alpha diversity metrics were analyzed using the Kruskal–Wallis test. Volatile fatty acids between LRFI and HRFI were compared using Student’s *t*-test for independent samples.

## 3. Results

### 3.1. Samples

In the calves, RFI values varied from −1.56 (lowest) to 4.76 (highest). One animal in the LRFI group could not be sampled (the steel collector tip did not enter the esophagus after several attempts) and we had to choose the animal with the next closest RFI value. The animal with the highest RFI value (4.76) from the HRFI group was not selected because the value was further than three standard deviations from the mean. Less efficient, HRFI animals, spent >20 min longer eating than LRFI animals, due to more visits per day not to the duration per feeding event (Table 1).

### 3.2. Sequencing Results

The sequencing procedure was successfully performed in all 40 DNA samples (10 calves, 2 time points, 2 DNA extraction methods). Method A yielded a lower DNA concentration and higher ratios of absorbance at 260/230 nanometers (nm), while the absorbances at 260/280 nm were similar between the two methods. A summary of the sequencing results is shown in Table 2.

### 3.3. Variation Analysis

To gain insights into the factors associated with the abundance in microbial groups, we calculated the variation between time points, DNA extraction methods, and RFI groups, using both the relative abundance of taxa and the clr-transformed data at the phylum level (see “Variation analysis” in Appendix A). A total of 20 taxa were discovered at the phylum level, but for this and other analyses, we removed two taxa because of low prevalence (phylum OD1 and 1 phylum from an unassigned kingdom). The removal of these two taxa allowed us to conduct the analysis of variance using data from 18 taxa at the phylum level (Appendix A). Except for Tenericutes, this analysis showed that the variation in microbial abundance was always higher between DNA extraction methods compared to the variation between days of sampling and between high- (LRFI) and low-efficiency (HRFI) animals.

### 3.4. Differences in Microbial Abundances between LRFI and HRFI

At the phylum level, there was no effect of the day of collection but there was an effect of the DNA extraction method across most microbial groups (*p* < 0.05). Three taxa showed a statistical difference (*p* < 0.05) between RFI levels (Chloroflexi, Elusimicrobia, and SR1). It is interesting that the DNA extraction method did not affect the abundance of all taxa (Bacteroidetes, Verrucomicrobia, TM7, and Tenericutes did not show a difference between DNA extraction methods, Table 3).

At the class level, there was no difference between days of sampling, but again most taxa were significantly different between DNA extraction methods (Appendix A). Anaerolineae (phylum Chloroflexi), Bacilli (Firmicutes), and an unassigned class of phylum SR1 were found to be significantly different between RFI levels (Appendix A). It is interesting to note that similar to the analysis of phyla, the strong effect of the DNA extraction method did not affect the abundance of all taxa at the class level.

The abundance of most taxa at the order level also showed significant differences between DNA extraction methods. Anaerolineales, Lactobacillales (class Bacilli), SR1, and RF32 (Proteobacteria) were found to be significantly different between RFI levels (Appendix A). It was interesting to note that there was no difference in Lactobacillales and other taxa between the two extraction methods. The order CW040 (phylum TM7, previously shown in ruminal contents of dairy heifers, [25]) again showed a trend for significance between RFI groups (*p* = 0.07) (Appendix A).

At the genus level, we detected 83 taxa (after filtering microbes with low abundance and prevalence). *Prevotella* (average: 28.7%), an unclassified member of the order Bacteroidales (6.7%) and the families Succinivibrionaceae (5.4%), Ruminococcaceae (5%) and Lachnospiraceae (4.8%) were the most abundant taxa accounting for ~50% of all microbial populations. There was no difference between sampling days and the abundance of most taxa was also different between DNA extraction methods (Appendix A). Most taxa did not show differences between LRFI and HRFI. Interesting exceptions include the propionate producer *Succiniclasticum* (family Veillonellaceae within the Firmicutes; note that the NCBI taxonomy database catalogs this taxon within the order Negativicutes) that was detected in all 40 samples with an average of 3.5% of all 16S sequences and lower values in LRFI animals (*p* = 0.0011) without the effect of extraction method. Other taxa that showed differences between LRFI and HRFI include the hemicellulose fermenter [26] BS11 (Bacteroidetes, detected in 36 of 40 samples with an average of 1%, higher in LRFI animals with *p* = 0.0021, and with a significant effect of extraction method), SHD-231 (family Anaerolinaceae, Chloroflexi, detected in 32 of 40 samples with an average of 0.07%, lower in LRFI animals with *p* = 0.0059, and with a significant effect of extraction method), RF32 (Alphaproteobacteria, detected in 33 of 40 samples with an average of 0.1%, higher in LRFI animals with *p* = 0.0069, and with a significant effect of extraction method), and others (Figure 1, Figure 2, Figure 3 and Figure 4, Appendix A). It is interesting to note that the differences between RFI groups also applied for relative abundances of *Succiniclasticum* (*p* = 0.0022, also without the effect of extraction method) and RF32 (*p* = 0.0399), but not for SHD-231 (*p* = 0.0911) and BS11 (*p* = 0.1062), although the residuals were not normally distributed for the last three, because of the presence of outliers.

### 3.5. Alpha Diversity

Alpha diversity refers to within-sample diversity and was calculated using four metrics. Samples from method A showed a higher number of features (*p* = 0.0002, Figure 5), evenness (*p* = 0.0005), faith (*p* = 0.06), and Shannon (*p* < 0.0001) compared to the results using method B. There was no significant difference between LRFI and HRFI in any of the metrics (Table 4 and Table 5). There was also no significant difference between days of sampling in any of the metrics.

### 3.6. Beta Diversity

Beta diversity refers to between-sample diversity and was calculated using four metrics. Using the filtered data, Bray–Curtis distances showed significant differences between DNA extraction methods (*p* = 0.001) and RFI (*p* = 0.002) levels (Figure 6). Additional comparisons showed that these differences between RFI levels applied to both method A (*p* = 0.023) and method B (*p* = 0.044). Similarly, Jaccard distances showed significant differences between DNA extraction methods (*p* = 0.001) and RFI (*p* = 0.002) levels. Additional comparisons showed that these differences between RFI levels applied to both method A (*p* = 0.026) and method B (*p* = 0.039). Unweighted UniFrac distances were significantly different between DNA extraction methods (*p* = 0.001) and almost reached significance for the comparison of LRFI and HRFI (*p* = 0.066), but additional comparisons between LRFI and HRFI within each method did not show differences (*p* > 0.1). Weighted UniFrac distances were also significantly different between methods (*p* = 0.001) and were not different between LRFI and HRFI (*p* = 0.107). Additional comparisons revealed a significant difference between LRFI and HRFI using samples from method A (*p* = 0.002) but not from method B (*p* = 0.312).

### 3.7. Functional Predictions Using PICRUSt2

Genes related to the superpathway of methylglyoxal degradation (*p* = 0.006) and polyamine biosynthesis (*p* = 0.006), and dTDP-N-acetylthomosamine biosynthesis (*p* = 0.009) were higher in LRFI. Other significant differences were found for genes related to L-lysine fermentation to acetate and butanoate (*p* < 0.001), pyruvate fermentation to acetone (*p* = 0.001), NAD salvage pathway (*p* = 0.002), arginine, ornithine, and proline interconversion (*p* = 0.002), glucose oxidative degradation (*p* = 0.004), succinate fermentation to butanoate (*p* = 0.008), 1,4-dihydroxy-2-naphthoate biosynthesis (*p* = 0.009), and phylloquinol biosynthesis (*p* = 0.009) (all lower in LRFI).

### 3.8. Volatile Fatty Acids

The concentration of VFAs in ruminal liquid samples was expressed as mM per L, converted to percentages, and analyzed using Student’s *t*-test. There was no significant difference in any fatty acid between LRFI and HRFI (*n* = 10 in each group, 5 animals with 2 sampling days, *p* > 0.18, Table 6). These results were not unexpected because of the well-known high rates of utilization and absorption of VFAs and the ability of multiple microbes to produce and use these compounds in vivo.

## 4. Discussion

In this study, the DNA extraction method proved to be key to delineating differences in ruminal microbes. The effect of DNA extraction is well-known in gut microbial ecology but is an important concern in studies of the rumen microbiome because many methods have been used in the literature depending on costs and availability, and even large studies of hundreds of animals have employed only one method [4]. Considering our results and other results about the effect of DNA extraction [10], we suggest, with reservations, employing more than one DNA extraction method and either combining the resulting DNA material to sequence only one sample (less expensive), or sequencing the DNA material from each method. This approach may provide a more comprehensive view of the actual rumen microbiome. Other authors have used only one DNA extraction method but have suggested even more complicated approaches, such as running small trials of several 16S regions to measure the discriminatory power of each region [27]. These and other suggestions may not only be unfeasible for some laboratories but may also prove to be inaccurate. For instance, we invite our colleagues to think about the criteria for objectively deciding which DNA extraction method (or 16S region) is better than others to truly reflect the microbial populations in their natural environment.

The taxa that show significant differences between animals with LRFI and HRFI are important for our understanding of feed efficiency, but care must be taken in the source of material for analysis because the liquid and solid fractions differ greatly in their microbial populations [8,28] and both are different from the bacteria attached to the rumen wall (epimural bacteria). Na and Guan [29] reviewed the role of rumen epithelial host–microbe interactions in cattle feed efficiency. In that study, the authors suggest at least three functions of epimural bacteria: tissue recycling, urea hydrolysis, and oxygen scavenging. Species of *Ruminococcus*, *Streptococcus*, *Prevotella*, and other bacteria in ruminal fluid have shown ureolytic activity in vitro [30] but it is challenging to prove this phenomenon in epimural bacteria. Moreover, even small differences in diets can promote differences or patterns of abundance in microbial taxa that could interfere with our understanding of microbial contributions to feed efficiency. For instance, *Streptococcus* in rumen fluid was correlated with RFI in 85 Braham steers on a low protein diet (8.8% crude protein) but not in a high protein diet (13.5% crude protein) [31], but this has not been investigated in epimural bacteria. This issue is not trivial since the choice of liquid samples responds to the ease and speed of taking samples and avoiding invasive fistulas.

*Succiniclasticum* is an interesting taxon that showed significantly lower abundance in high-efficiency, LRFI animals, without a DNA extraction effect. It was first described in 1995 as a small, rod-shaped ruminal bacterium capable of converting succinate to propionate as the sole energy-yielding mechanism [32]. The isolated microbe did not ferment carbohydrates, produce urease, or reduce nitrate, and depended on rumen fluid and yeast extract for good growth [32]. The same author with other colleagues later showed that *Schwartzia succinivorans* was another ruminal bacterium utilizing succinate as the sole energy source that also depended on rumen fluid and yeast extract [33] but in this study, we did not detect differences in this taxon (Appendix A). Interestingly, Myer et al. [34] also showed that *Succiniclasticum* was detected at the greatest abundance in low-efficiency, mixed breeds steers, more specifically in animals in the subgroup ADG_Low_-ADFI_High_. *Succiniclasticum* was found in greater proportion in the liquid fraction of beef steers on a low-quality forage diet [35] and was mostly undetected on forage diets but was more abundant in a high-grain diet [36]. Another study showed that *Succiniclasticum* was the most abundant taxon (9.4%, compared to 3.5% in this study) in rumen liquid from slaughtered finishing bulls [37], and Petri et al. [38] showed that this and other taxa were particularly prevalent during ruminal acidosis in the rumen epimural microbiota. One study showed that feed restriction was associated with a large reduction in an uncharacterized Succinivibrionaceae species (OTU-S3004), with important differences between the liquid and solid fraction [39], and Luo et al. [40] showed that a high-concentrate diet (forage-to-concentrate ratio = 20:80) plus niacin, increased the abundance of *Succiniclasticum* and other taxa in ruminal fluid from cannulated Jinjiang cattle. Since different fractions of ruminal contents differ greatly in their microbial populations [8,28], future studies should consider looking at both the solid and the liquid fractions, as well as the rumen wall, to better understand the relationship between feed efficiency and this taxon.

The question of why *Succiniclasticum* is lower in highly efficient animals deserves attention. This taxon was present in all 40 samples and at high abundance (3.5%), which suggests that it is a member of the core rumen microbiome of beef cattle (another study suggested that this also applies to other ruminant animal species, see [41]). To explain the lower abundance of *Succiniclasticum* in efficient animals, Myer et al. [34] suggested a phenomenon of resource competition because several members of another propionate producer (unknown member of the family Veillonellaceae) were decreased in the same ADG_Low_-ADFI_High_ subgroup. While this hypothesis is feasible, the rationale behind the belief in a relationship between the abundance of propionate producers and feed efficiency is debatable. For instance, it is also possible to think of a rumen ecosystem with more efficient microbes that require fewer numbers to produce the same or more propionate (in this regard, interesting new concepts such as modularity are now being applied in ruminal microbial ecology, [42]). It is also possible that the oxaloacetate is used in other, anabolic routes, rather than serving for succinate formation [43]. On the other hand, grass-fed cattle have lower propionate concentrations compared to grain-fed cattle [42], but this phenomenon may relate to the digestibility of the feed, ruminal passage rate, and the sampling time after the last meal. Moreover, it is believed that the solids of rumen contents are the “chief” substrate for succinate production [44] but here we did not investigate the presence of microbes in solid contents. The production of propionate from succinate is also linked to H_2_ concentrations which regulate the thermodynamics of rumen fermentation [43,45] that in turn serve to regulate microbial abundances. Finally, the original observation of dependence on yeast extract [32] is worth looking at.

The hemicellulose fermenter BS11 [26] from Bacteroidetes was detected in most samples with an average of 1% and showed a higher abundance in LRFI animals. It is interesting that this taxon was found in higher abundance in conditions where there was a need for more efficiency (enriched in winter diets in Alaskan moose, [26]). Chloroflexi is another interesting taxon that is commonly found in treatment plants where they feed on lysed bacterial cells and degrade complex organic compounds [46]. Representatives of this taxon have also been found in the human and animal gut microbiota [47,48], including ruminal fluid from dairy heifers [49]. In this study, the genus SHD-231 (family Anaerolinaceae, Chloroflexi) was affected by both the DNA extraction method and RFI level (lower in LRFI animals). Members of the family Anaerolinaceae are common in anaerobic digesters [50] but the relevance of this taxon in ruminal microbial ecology and feed efficiency remains unknown.

Without measures of gene expression, the predictions suggested by PICRUSt are usually not very informative, but some features may deserve attention. Genes related to the superpathway of methylglyoxal degradation were higher in LRFI. Methylglyoxal is produced by ruminal bacteria in response to low nitrogen levels in the rumen [51]. Methylglyoxal is a highly toxic, alternative end product of glucose fermentation that is produced by bacteria when there is carbohydrate excess, and nitrogen limitation and can kill cells and inhibit protein synthesis [52,53]. A gene involved in methylglyoxal degradation, lactoylglutathione lyase (glo1), has been suggested as a strong candidate biomarker of rumen microbiome in less efficient animals because of its higher relative abundance in a shotgun metagenomic sequencing study [54]. Interestingly, Petri et al. [55] showed that carbohydrate metabolism based on the glyoxylate pathway is increased in correlation with *Succiniclasticum*, adding another valuable piece of information about this microbe.

The limitations of this study include the low number of animals at each extreme of RFI values, as larger studies often result in more precise and useful results for the scientific community and the beef cattle industry. The inclusion of more breeds can also be something to consider in future studies because breed is strongly associated with feed efficiency [3]. Also, the diet used in this study is not the diet used in commercial feedlots, and it remains controversial whether the feed efficiency results can be extrapolated. Moreover, this and other studies about feed efficiency in cattle have analyzed the rumen of the microbiota of males only, and if the microbiota can be inherited [4], then the relevance of microbial inheritance is questionable because the outspring destined for beef cattle fattening operations have no contact with the microbiota from the father, or even with their real mothers in females that have been inseminated. Finally, the collection of samples from other anatomical sites (e.g., the lower gut) can provide a more complete understanding of the complex relationship between the gut microbiota and feed efficiency [56,57], and other molecular techniques such as quantitative real-time PCR could help to confirm the relative abundance and variation in microbial taxa.

In conclusion, this paper shows a strong effect of the DNA extraction method in describing ruminal microbial communities and the existence of several microbes that could be related to feed efficiency based on RFI calculations, such as *Succiniclasticum*, and members of Bacteroidetes and Chloroflexi. It is interesting to note that all these groups were not necessarily the most abundant, which fits with the idea that low-abundance microbes display complex interactions that help maintain community stability [58]. The implications of these findings in beef cattle operations include a more precise understanding of the key microbial players affecting feed efficiency, which may result in more useful strategies to help produce more and better meat with the same amount or less feed.

## Figures and Tables

**Figure 1 microorganisms-11-00663-f001:**
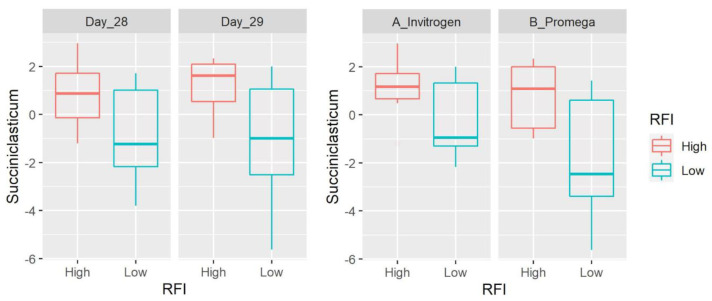
Box plots showing the abundance of *Succiniclasticum* (Firmicutes) based on clr-transformed data. Data are shown for both day of sampling and DNA extraction method. There was a significant difference between LRFI and HRFI (*p* = 0.0011), with no time or DNA extraction effect.

**Figure 2 microorganisms-11-00663-f002:**
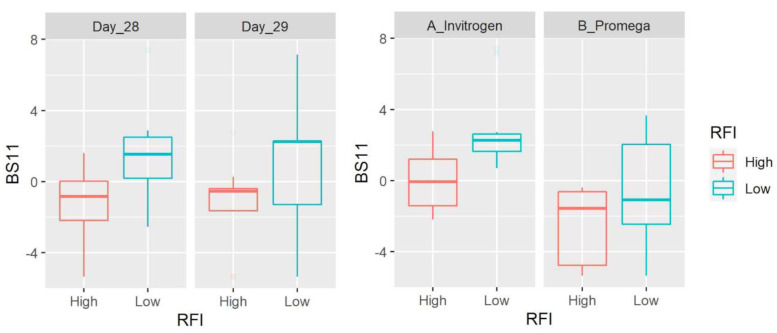
Box plots showing the abundance of BS11 (Bacteroidetes) based on clr-transformed data. Data are shown for both day of sampling and DNA extraction method. There was a significant difference between LRFI and HRFI (*p* = 0.0021) and DNA extraction methods (*p* = 0.0003), with no time effect.

**Figure 3 microorganisms-11-00663-f003:**
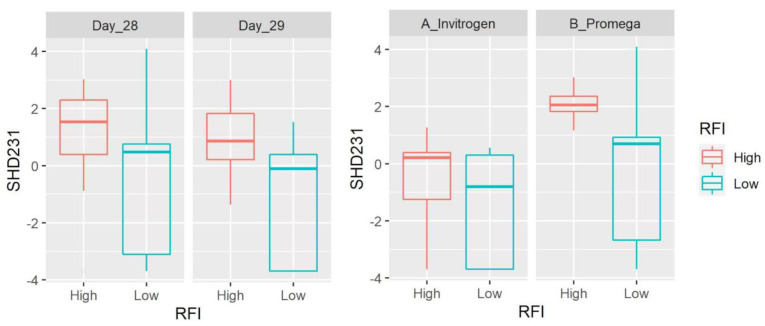
Box plots showing the abundance of SHD-231 (family Anaerolinaceae, Chloroflexi) based on clr-transformed data. Data are shown for both day of sampling and DNA extraction method. There was a significant difference between LRFI and HRFI (*p* = 0.0059) and DNA extraction methods (*p* = 0.0020), with no time effect.

**Figure 4 microorganisms-11-00663-f004:**
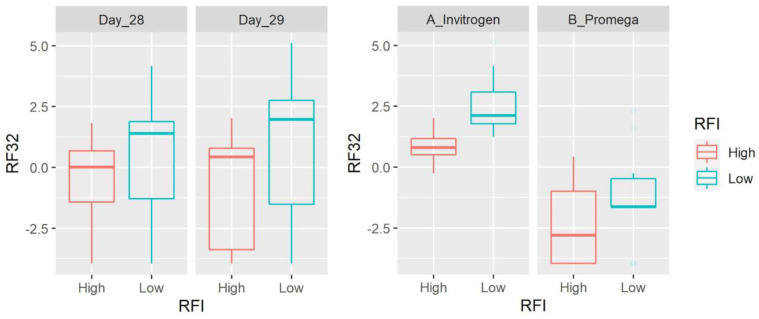
Box plots showing the abundance of RF-32 (Alphaproteobacteria) based on clr-transformed data. Data are shown for both day of sampling and DNA extraction method. There was a significant difference between LRFI and HRFI (*p* = 0.0069) and DNA extraction methods (*p* < 0.0001), with no time effect.

**Figure 5 microorganisms-11-00663-f005:**
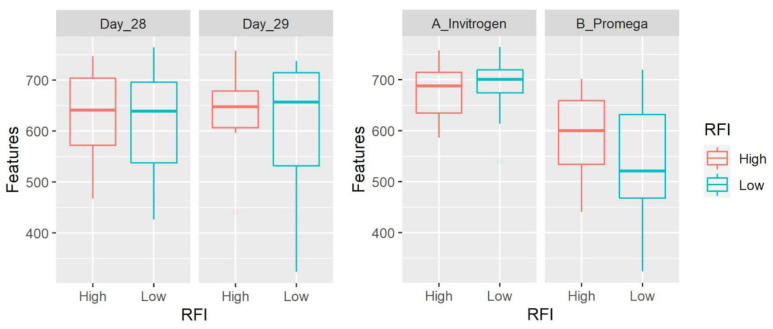
Box plots showing number of features. Data are shown for both day of sampling and DNA extraction method. The number of features and all other alpha diversity metrics were significantly higher using samples from method A with no significant differences between LRFI and HRFI.

**Figure 6 microorganisms-11-00663-f006:**
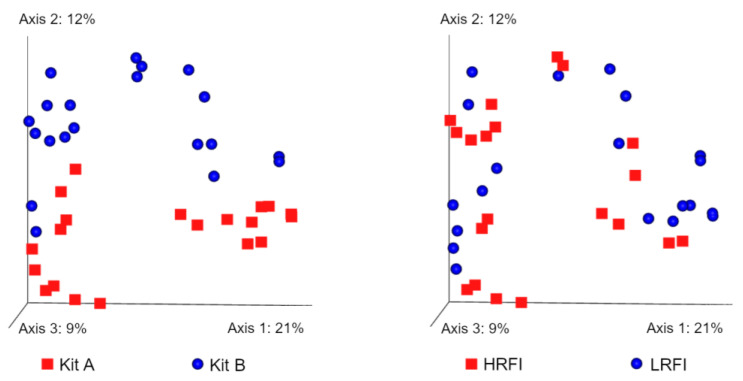
Principal coordinate analysis (PCoA) plots showing Bray–Curtis distances. The separation of samples based on the DNA extraction method is much more evident compared to the separation of samples based on RFI levels. The strong effect of DNA extraction method was also observed in other beta diversity metrics.

**Table 1 microorganisms-11-00663-t001:** Descriptive statistics for productive parameters for LRFI and HRFI ^1^.

Item	Efficient (LRFI)	Inefficient (HRFI)	*p*-Value
RFI	−1.17 (−0.74, −1.56)	0.89 (0.65, 1.29)	0.008
Start weight (kg)	370.4 (344.6, 387.5)	312.9 (235.4, 397.6)	0.22
End weight (kg)	491.3 (461.5, 528.4)	437.8 (343.2, 552.4)	0.40
Duration per feeding event (seconds)	199.8 (170.9, 249.9)	197.8 (138.0, 256.7)	0.94
Intake per feeding event (grams)	307.7 (247.7, 353.0)	281.1 (152.5, 363.5)	0.57
Visits per day	30.3 (26.3, 33.8)	41.2 (29.1, 63.9)	0.21
Feeding duration per day (minutes)	101.0 (85.5, 135.6)	128.1 (110.9, 147.1)	0.06
ADG (kg/day) ^2^	1.57 (1.35, 1.60)	1.62 (1.21, 2.01)	0.53
DMI (kg/day) ^3^	8.77 (7.54, 10.02)	10.12 (7.87, 12.83)	0.21
Raw F:G	5.64 (5.31, 6.51)	6.33 (5.62, 8.24)	0.29
Adj. F:G ^4^	5.51 (4.46, 5.04)	6.62 (5.04, 8.16)	0.21

^1^ Average (minimum, maximum) are provided. ^2^ Average daily gain. ^3^ Dry matter intake. ^4^ Adjusted feed-to-gain ratio (F:G) accounts for differences in animal age and size during tests. The values are the base F:G values multiplied by the trial group’s metabolic mid-weight divided by the individual’s metabolic mid-weight.

**Table 2 microorganisms-11-00663-t002:** Sequencing results for each DNA extraction method ^1^.

Method	Raw Input Sequences	Sequences after Filtering	Non-Chimeric
A	1,501,305	1,491,595	796,559
B	1,578,049	1,547,278	802,904

^1^ A total of 20 samples were processed with each DNA extraction method, 10 calves from 2 time points.

**Table 3 microorganisms-11-00663-t003:** Summary of statistical results (*p*-values) at the phylum level ^1^.

Phylum	Time Points	DNA Extraction Methods	RFI Groups
Actinobacteria	0.1520	<0.0001	0.1449
Bacteroidetes ^2^	0.7356 NP (*p* = NS)	0.1503 NP (*p* = NS)	0.8797 NP (*p* = NS)
Chloroflexi	0.4963	0.0020	0.0059
Cyanobacteria	0.6211	0.0002	0.3826
Elusimicrobia	0.5500	<0.0001	0.0157
Euryarchaeota	0.2068	<0.0001	0.2544
Fibrobacteres	0.3888	<0.0001	0.9421
Firmicutes	0.5853	0.0007	0.3500
Lentisphaerae	0.4119	0.0579	0.8404
Planctomycetes	0.2967	<0.0001	0.6573
Proteobacteria	0.1577	0.0008	0.3968
Spirochaetes	0.4740	<0.0001	0.1146
SR1	0.6005	<0.0001	0.0338
Synergistetes	0.8530	0.0369	0.2161
Tenericutes	0.1650	0.7420	0.3420
TM7	0.7447	0.1730	0.0731
Unassigned phylum ^2^	0.8166 NP (*p* = NS)	0.0012 NP (*p* = NS)	0.4094 NP (*p* = NS)
Verrucomicrobia	0.4860	0.5238	0.6398

^1^ We used PROC MIXED in SAS University Edition using the clr-transformed data from each taxon as the dependent variable, and day of sampling, DNA extraction method, and RFI, as independent variables, without random effects. ^2^ Residuals not normally distributed. Four taxa (Bacteroidetes, Verrucomicrobia, TM7, and Tenericutes) did not show a difference between DNA extraction methods. NP: non-parametric analysis, NS: non-significant (*p* > 0.05).

**Table 4 microorganisms-11-00663-t004:** Mean alpha diversity parameters from method A using the filtered table.

Parameter	LRFI	HRFI	*p*-Value ^1^
Observed features	685	680	0.65
Evenness	0.79	0.81	0.41
Faith PD	30.3	28.7	0.23
Shannon	7.5	7.6	0.65

^1^ *p*-values come from the Kruskal–Wallis test. A minimum of 10,000 sequences was chosen to include all samples. A filtered table refers to a table where features appearing in less than 4 samples and with less than 20 in frequency were removed.

**Table 5 microorganisms-11-00663-t005:** Mean alpha diversity parameters from method B using the filtered table.

Parameter	LRFI	HRFI	*p*-Value ^1^
Observed features	539	587	0.36
Evenness	0.77	0.78	0.41
Faith PD	26.9	27.6	0.65
Shannon	6.9	7.2	0.49

^1^ *p*-values come from the Kruskal–Wallis test. A minimum of 10,000 sequences was chosen to include all samples. A filtered table refers to a table where features appearing in less than 4 samples and with less than 20 in frequency were removed.

**Table 6 microorganisms-11-00663-t006:** Volatile fatty acids (VFAs) for LRFI and HRFI ^1^.

Item	Efficient (LRFI)	Inefficient (HRFI)	*p*-Value
Acetic (mM/L)	38.8 (28.6, 47.9)	36.9 (23.7, 51.2)	0.61
Propionic (mM/L)	14.9 (6.6, 22.2)	12.8 (4.5, 21.2)	0.44
Butyric (mM/L)	6.4 (4.2, 9.3)	6.6 (3.7, 12.2)	0.84
Total VFAs	60.1 (41.6, 78.4)	56.3 (33.1, 81.7)	0.58
Acetic (%)	64.9 (59.3, 72.4)	66.7 (59.6, 75.1)	0.47
Propionic (%)	24.4 (15.9, 31.7)	21.6 (13.7, 30.2)	0.32
Butyric (%)	10.7 (7.8, 13.7)	11.7 (10.2, 15.1)	0.18

^1^ Average (minimum, maximum) are provided.

## Data Availability

The data used to support the findings of this study are available from the first author upon request.

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
