# Peer review of "Microbial Populations in Ruminal Liquid Samples from Young Beefmaster Bulls at Both Extremes of RFI Values"

_microorganisms, 2023, doi:10.3390/microorganisms11030663_

Round 1

Reviewer 1 Report

The ruminal samples were obtained from Beefmaster steers with LRFI and HRFI to detect the

The author detected the difference of rumen microbiota between LRFI and HRFI cattle by 16S rRNA in V3 and V4 regions. In addition, the effect of DNA extraction methods on the rumen microbiota diversity and abundance was also detected. The results showed that the abundance of some ruminal microbes is associated with feed efficiency and serve as a cautionary tale for the interpretation of results obtained with a single DNA extraction method. This study has certain guiding significance for accurate detection and analysis of animal gut microbiota.

1.      How is rumen fluid collected? How to deal with it? Is it filtered?

2.      The microbiological analysis method is too simple.

3.      The result is too simplistic. How many phylum levels are detected? How many genus levels? What are the main phyla? What are the main bacteria?

4.      The extraction method affected the structure of rumen microflora. Were the bacteria with high abundance or low abundance affected? The bacteria that are different between LRFI and HRFI? The elaboration of these results has important reference significance for summarizing the effects of extraction methods on rumen microorganisms.

Author Response

Please find below our responses to your queries and suggestions, and attached is the whole set of responses to all reviewers.

REVIEWER 1

The ruminal samples were obtained from Beefmaster steers with LRFI and HRFI to detect the

RESPONSE: This sentence seems incomplete. We would be happy to address any comment or suggestion from Reviewer 1. Please find below our responses to your valuable queries.

The author detected the difference of rumen microbiota between LRFI and HRFI cattle by 16S rRNA in V3 and V4 regions. In addition, the effect of DNA extraction methods on the rumen microbiota diversity and abundance was also detected. The results showed that the abundance of some ruminal microbes is associated with feed efficiency and serve as a cautionary tale for the interpretation of results obtained with a single DNA extraction method. This study has certain guiding significance for accurate detection and analysis of animal gut microbiota.

RESPONSE: We appreciate your comments about the significance of our research work.

  1. How is rumen fluid collected? How to deal with it? Is it filtered?

RESPONSE: As mentioned in the main text, we used SELEKT equipment (https://www.nimrodvet.com/selekt-ruminants-equipment/). The steel collector tip does not allow the passage of solids, which can clog the hose. Based on our experience and the current literature, we do not recommend filtering the ruminal liquid because this may discard some microbial populations. The section has been updated.

  1. The microbiological analysis method is too simple.

RESPONSE: We used the most sophisticated molecular techniques and bioinformatics tools but would be happy to explore additional options based on the reviewer’s suggestions.

  1. The result is too simplistic. How many phylum levels are detected? How many genus levels? What are the main phyla? What are the main bacteria?

RESPONSE: We are curious about what section of results specifically is too simplistic. The number of taxa at the phylum level were stated in the results. We also added the number of taxa at the class (Supplementary Table S2), order (Supplementary Table S3) and genus (Supplementary Table S4) level. While the “main” bacteria is not necessarily the most abundant (DOI:10.1371/journal.pcbi.1005361), we updated the results on the main manuscript at the genus level.

  1. The extraction method affected the structure of rumen microflora. Were the bacteria with high abundance or low abundance affected? The bacteria that are different between LRFI and HRFI? The elaboration of these results has important reference significance for summarizing the effects of extraction methods on rumen microorganisms.

RESPONSE: We thank the reviewer for sharing these very important thoughts. It is very interesting (we did not think about this before reading your thoughts) that all groups that showed differences between RFI levels were not necessarily the most abundant, which fits with the idea that low-abundant microbes display complex interactions that help maintain community stability (DOI:10.1371/journal.pcbi.1005361). The conclusion paragraph was updated based on your suggestion.

Reviewer 2 Report

This research is very important method research, so it's strong novelty, but there is still some problems, as follows:

In introduction, please detail describe the progress of detecting methodological fators' good and bad characteristics, so please authors supplement this progress.

In material and method, the experimental details is not clear, specially about critical steps. Because this method research, so this side is very important.

In results, in some table, there is no units, for example, Table 1 the ADG and DMI should be supplemented g/day or kg/kg, please check all table contents.

In disscussion, please supplement more disscussion and more references, this part is less, so please reviese this part.

In conclusion, please authors give one more confirmed results and some suggests for others' researchers, make some alerts in similar research.

To sum up, this research is very novelty and useful, if authors make revision according above suggestion, this manuscript should be published in this journal.

Author Response

Please find below our responses to your queries and suggestions, and attached the full set of responses to all reviewers.

REVIEWER 2

This research is very important method research, so it's strong novelty, but there is still some problems, as follows:

RESPONSE: We appreciate your comments about the significance of our research work. Please find below our responses to your queries and suggestions.

In introduction, please detail describe the progress of detecting methodological factors' good and bad characteristics, so please authors supplement this progress.

RESPONSE: The variations among methodologies are important because they can provide different views of the structure and composition of microbial populations, which are not necessarily good or bad. The last paragraph in the introduction was updated based on your query.

In material and method, the experimental details is not clear, specially about critical steps. Because this method research, so this side is very important.

RESPONSE: The materials and methods section, particularly the collection of ruminal samples, has been updated.

In results, in some table, there is no units, for example, Table 1 the ADG and DMI should be supplemented g/day or kg/kg, please check all table contents.

RESPONSE: Table 1 has been updated (ADG and DMI are expressed as kg/day). All other tables are complete.

In discussion, please supplement more discussion and more references, this part is less, so please revise this part.

RESPONSE: The discussion contain key aspects of our results, based on the most current literature, but the last area of discussion has been expanded to better stimulate future research work.

In conclusion, please authors give one more confirmed results and some suggests for others' researchers, make some alerts in similar research.

RESPONSE: The conclusion has been updated to better stimulate future research work.

To sum up, this research is very novelty and useful, if authors make revision according above suggestion, this manuscript should be published in this journal.

RESPONSE: We very much appreciate your comments about the significance and novelty of our research work.

Reviewer 3 Report

There are many repetitive sentences across the manuscript and needs to be largely reworked. 

Introduction

The main objective should be coupled with the hypothesis of the study and should be clearly stated in the introduction.

Material and Methods

The methods information should be adequately provided. The Ethical clearance (code number) of the study is missing. The selection criteria of Beefmaster steers and calves should be described in details, e.g., age, body weight, gender etc.? Additionally, the variation between individual animals should be assessed, or at least taken into consideration. Also, the human handlers can affect the structure of microbiota. These effects must be excluded before considering any significant difference. The method of sample preparation (e.g., equal weight of the rumen liquor sample from each animal being pooled) must be provided. Were the specimen pooled in prior to DNA extraction?

Discussion

The discussion section needs to be revised in order to improve the quality of the Manuscript / include the following points in which authors need to address thoroughly:

1.       How different are the methods for DNA extraction implemented by the authors in their study than what has been reported in the literature? Such valuable information is missing in the discussion.

2.       Which of the resident microbiota aids in determining the ability of the host to harvest energy from its diet pertaining to analysis of VFAs concentrations.

3.       Assess the impact of microbial populations on feed efficiency.

4.       The intestinal bacteria should also be considered when examining the basis of ruminant feed efficiency.

5.       Observe any correlations between animals with LRFI and HRFI and ruminal bacterial families. 

Author Response

Please find below our responses to your queries and suggestions, and attached the full set of responses to all reviewers.

REVIEWER 3

There are many repetitive sentences across the manuscript and needs to be largely reworked.

RESPONSE: The manuscript has been reworked considering your valuable comments and suggestions as well as the input from other 4 reviewers.

Introduction

The main objective should be coupled with the hypothesis of the study and should be clearly stated in the introduction.

RESPONSE: The objective of this research work was clearly stated in the last sentence of the introduction but this last section of the introduction has been updated with regards to the relevance of the variations among methodologies to provide a more comprehensive view of microbial populations.

Material and Methods

The methods information should be adequately provided. The Ethical clearance (code number) of the study is missing.

RESPONSE: We section on material and methods has been expanded with more details and the protocol number was included in the first draft of the manuscript.

The selection criteria of Beefmaster steers and calves should be described in details, e.g., age, body weight, gender etc.? Additionally, the variation between individual animals should be assessed, or at least taken into consideration. Also, the human handlers can affect the structure of microbiota. These effects must be excluded before considering any significant difference. The method of sample preparation (e.g., equal weight of the rumen liquor sample from each animal being pooled) must be provided. Were the specimen pooled in prior to DNA extraction?

RESPONSE: As mentioned in the main text, in this research work we selected males only from a feed efficiency test, in which the animals are selected by the association (in this case, Beefmaster). We agree that the handlers may have an effect on the behavior of the animals but all animals were treated similarly throughout the entire duration of the test. We collected 30 mL of liquid fluid from each animal at each sampling day and were analyzed separately to consider inter-day variation in the microbiota. The section has been updated. Please do note that 4 different aliquots of 1.5 mL each (2 for each DNA extraction kit) were processed and the pellets in each set of 2 tubes were mixed in one tube which was the source material for each DNA extraction procedure. These additional steps are not common in metagenomic studies (including our own previous research studies) but were performed here in an effort to provide a more comprehensive view of the microbial populations in the ruminal liquid samples.

Discussion

The discussion section needs to be revised in order to improve the quality of the Manuscript / include the following points in which authors need to address thoroughly:

  1. How different are the methods for DNA extraction implemented by the authors in their study than what has been reported in the literature? Such valuable information is missing in the discussion.

RESPONSE: This is a very important query. Many different methods have been used in the literature but, in our experience, researchers do not choose one over the others because of a more precise view of microbial communities, but because of costs and availability in their countries. The discussion has been updated on this regard:

The effect of DNA extraction is a well-known phenomenon in gut microbial ecology but is an important concern in studies of the rumen microbiome because many methods have been used in the literature depending on costs and availability, and even large studies of hundreds of animals have employed only one method [4]. Considering our results and other results about the effect of DNA extraction [10], we suggest, with reservations, employing more than one DNA extraction method and either combine the resulting DNA material to sequence only one sample (less expensive), or sequence the DNA material from each method. This approach may provide a more comprehensive view of the actual rumen microbiome.

  1. Which of the resident microbiota aids in determining the ability of the host to harvest energy from its diet pertaining to analysis of VFAs concentrations.

RESPONSE: This is very difficult to prove due to the ability of multiple taxa to produce VFAs and the large number of uncultivated members of the rumen microbiome. That is why we decided to focus on the microbes in the discussion. However, the results section about VFAs was expanded based on your valuable query.

  1. Assess the impact of microbial populations on feed efficiency.

RESPONSE: We would very much like to do this but it is also important to be cautious, particularly because our results are based on 16S gene sequencing. However, we used your thoughts and the queries from other reviewers to expand the conclusion section.

  1. The intestinal bacteria should also be considered when examining the basis of ruminant feed efficiency.

RESPONSE: We agree, that is why we included this important information in the section of limitations.

  1. Observe any correlations between animals with LRFI and HRFI and ruminal bacterial families.

RESPONSE: Thank you, before submission we performed correlation analyses between clr-transformed data and RFI values, but every individual RFI value (from each of the 10 steers) is associated with 4 different clr-transformed data values. This creates a pattern that does not allow a meaningful correlation analysis (correlation analysis is best when there are different values on x and y and both are evenly distributed along their respective axis).

Reviewer 4 Report

Dear authors, it seems to me that this is a well written manuscript and it has relevance in the scientific world. However, some points affect the quality of the manuscript.

 General comments:

Abstract:

The abstract is very generic. Try adding data about your results. Add the objective of the study in the abstract and if possible the conclusion.

Introduction:

The introduction is well written; however, why did you choose two extraction methods and not 3 or 4? Why these two extraction methods? If RFI can modulate the rumen microbiome, how can you know if the results obtained with method A (or B) are correct? Add all the information about this in the introduction.

Material and methods:

Describe the complete method for obtaining rumen fluid, post-harvest treatment and storage. Because this could be a methodological error of the study.

Discussion:

The discussion is a very well put together review. I like it and probably the readers will like it. However, the function of a discussion is to explain your results. In this sense, add paragraphs that explain each result obtained through theories, hypotheses or facts; according to the results.

Specific comments:

Lines 17-18: These two ideas are not connected. The microbiota is involved in productivity and the different analysis strategies may not be similar. But what is the connection between these ideas?

Lines 23-24: These lines are very generic. Try to be more specific. What do you mean by “taxa”? A new reader may not know the terminology. The results can focus on the most important species of bacteria instead of giving a generic answer that is not generic.

Lines 25-26: What do you mean by diversity measures and functional predictions? I understand the terms but no previous ideas were introduced.

Lines 28-30: These lines are not correlated with the results and the objective.

Line 57: delete "as discussed above"

Lines 73-74: The total of the ingredients is not 100%.

Line 83: Try to describe this part more specifically and clarify the two DNA extraction methods used, because this is one of the objectives of the study.

Line 158: Eliminate "Other productive... Table 1"

Line 175: Time points? This is not described in the aim of the manuscript.

Line 166: Perhaps you can combine this subtopic with subtopic 3.3.

Line 187: Were the sampling days also a studied item? Why isn't this explained before?

Line 206: On line 201-202 you explain what it affects and on this line it does not affect. It is very contradictory. You should avoid contradictory ideas because that lowers the quality of your ability to argue.

Lines 262-264: Were the teams also part of the study? If the kit is the extraction method, describe it as Method A and Method B.

Lines 345-347: Why 2 methods? Explain at this point why two methods. If no differences are found; Why two methods? If differences were found. Combining methods with different results can create a different version of the actual rumen microbiome situation. From that point of view, do you think your suggestion is correct?

Conclusion: In this sense, did only the extraction method show differences in the rumen microbiome community? RFI did not promote differences? I think you need to rewrite the conclusion considering your results. Increase the implications part, because this is an important topic.

Author Response

Please find below our responses to your queries and suggestions, and attached the full set of responses to all reviewers.

REVIEWER 4

Dear authors, it seems to me that this is a well written manuscript and it has relevance in the scientific world. However, some points affect the quality of the manuscript.

RESPONSE: We appreciate your comments about the relevance of our research work. Please find below our responses to your queries and suggestions.

 General comments:

Abstract:

The abstract is very generic. Try adding data about your results. Add the objective of the study in the abstract and if possible the conclusion.

RESPONSE: Thank you, this was also pointed out by another reviewer. However, the word count is limited and only allowed us to rephrase key sentences.

Introduction:

The introduction is well written; however, why did you choose two extraction methods and not 3 or 4? Why these two extraction methods? If RFI can modulate the rumen microbiome, how can you know if the results obtained with method A (or B) are correct? Add all the information about this in the introduction.

RESPONSE: We agree and appreciate these important queries. Unless you have a method that is independent of the effect of DNA extraction method, nobody can say if one DNA extraction method is “better” than others to truly reflect the microbial populations in any microbial environment. That is why we discuss this in the second paragraph of the discussion.

There is no specific reason why we chose those 2 DNA extraction methods, it was simply a matter of availability. We rephrased this in the introduction.

Material and methods:

Describe the complete method for obtaining rumen fluid, post-harvest treatment and storage. Because this could be a methodological error of the study.

RESPONSE: This section has been updated as follows:

At day 5 after the end of the evaluation period, we obtained the RFI values from a GrowSafe system [15] and ruminal liquid samples were obtained from 10 steers at both extremes of RFI values (5 with the lowest and 5 with the highest RFI) from two consecutive days (Day 1 and Day 2, to consider inter-day variation) using SELEKT equipment (Nimrod Veterinary Products Ltd, Moreton-in-Marsh, Gloucestershire, UK). The steel collector tip in this equipment does not allow the passage of solids, which can clog the hose, and samples were not filtered because this may discard some microbial populations. Ruminal samples (~30 mL) were placed in a 50 mL tube containing 10 mL of ethanol at 95% for better microbiome stability [9] and frozen at -20°C until DNA extraction and volatile fatty acids (VFA) analysis.

Discussion:

The discussion is a very well put together review. I like it and probably the readers will like it. However, the function of a discussion is to explain your results. In this sense, add paragraphs that explain each result obtained through theories, hypotheses or facts; according to the results.

RESPONSE: Thank you for your comments on the way we organized the discussion. Key areas in the discussion have been updated using all reviewers’ comments and suggestions.

Specific comments:

Lines 17-18: These two ideas are not connected. The microbiota is involved in productivity and the different analysis strategies may not be similar. But what is the connection between these ideas?

RESPONSE: The two ideas have been connected as follows:

The gut microbiota is involved in productivity of beef cattle, but the impact of different analysis strategies on microbial composition is unclear.

Lines 23-24: These lines are very generic. Try to be more specific. What do you mean by “taxa”? A new reader may not know the terminology. The results can focus on the most important species of bacteria instead of giving a generic answer that is not generic.

RESPONSE: The word count in the abstract is limited but we did change the word “taxa” for the word “microbes”.

Lines 25-26: What do you mean by diversity measures and functional predictions? I understand the terms but no previous ideas were introduced.

RESPONSE: The word count in the abstract is limited with no room for previous ideas. We hope that unfamiliar readers can use the contents in the main text to better understand the terms and thoughts.

Lines 28-30: These lines are not correlated with the results and the objective.

RESPONSE: The word count in the abstract is limited but we believe that the last sentence of the abstract is well connected with the objective and results in this paper.

Line 57: delete "as discussed above"

RESPONSE: Deleted, thanks.

Lines 73-74: The total of the ingredients is not 100%.

RESPONSE:  Based on your comment and the comments from other reviewers, we decided to add the full list of ingredients as Supplementary Information.

Line 83: Try to describe this part more specifically and clarify the two DNA extraction methods used, because this is one of the objectives of the study.

RESPONSE: Thank you for pointing this put. The material and methods section and subsection 3.2 of results have been updated.

Line 158: Eliminate "Other productive... Table 1"

RESPONSE: Deleted.

Line 175: Time points? This is not described in the aim of the manuscript.

RESPONSE: We collected samples from two consecutive days to consider the well-known inter-day variation in gut microbiota as described in material and methods.

Line 166: Perhaps you can combine this subtopic with subtopic 3.3.

RESPONSE: Since they refer to different topics, we believe that keeping them separated may aid the flow of the manuscript.

Line 187: Were the sampling days also a studied item? Why isn't this explained before?

RESPONSE: We decided to obtain samples from two consecutive days because of the well-known inter-day variation in the gut microbiota, as stated in material and methods. We do not consider this to be a studied item as such.

Line 206: On line 201-202 you explain what it affects and on this line it does not affect. It is very contradictory. You should avoid contradictory ideas because that lowers the quality of your ability to argue.

RESPONSE: The sentence “most taxa were significantly different between DNA extraction methods” does not contradict the sentence “DNA extraction method did not affect the abundance of all taxa” but the text has been updated to make it more understandable.

Lines 262-264: Were the teams also part of the study? If the kit is the extraction method, describe it as Method A and Method B.

RESPONSE: All areas with either kit A or kit B were changed to method A or method B, respectively.

Lines 345-347: Why 2 methods? Explain at this point why two methods. If no differences are found; Why two methods? If differences were found. Combining methods with different results can create a different version of the actual rumen microbiome situation. From that point of view, do you think your suggestion is correct?

RESPONSE: Very good point. From a perspective of the technicians performing DNA extraction procedures in the laboratory, it would be unfeasible to use more than 2 methods. That is why we said: “with reservations, employing at least 2 methods”. The section was rewritten as follows:

Considering our results and other results about the effect of DNA extraction [10], we suggest, with reservations, employing more than one DNA extraction method and either combine the resulting DNA material to sequence only one sample (less expensive), or sequence the DNA material from each method. This approach may provide a more comprehensive view of the actual rumen microbiome.

Conclusion: In this sense, did only the extraction method show differences in the rumen microbiome community? RFI did not promote differences? I think you need to rewrite the conclusion considering your results. Increase the implications part, because this is an important topic.

RESPONSE: Thank you for helping us out improve the implications part. The sentence was rewritten as follows:

The implications of these findings in beef cattle operations include a more precise understanding of the key microbial players affecting feed efficiency, which may result in more useful strategies to help produce more and better meat with the same or less feed.

Reviewer 5 Report

attached comments

Author Response

Please find below our responses to your queries and suggestions, and attached the full set of responses to all reviewers.

REVIEWER 5

Thank you for your time in revising our research work. Please find below our responses to your valuable queries and suggestions.

ABSTRACT

Sample size: As most of our colleagues studying gut microbial ecology, we did not perform and do not recommend performing power analysis, mainly because the effect size is shared among hundreds of microbial taxa. A good classic example of this is an exploration of the effects of an antibiotic on the fecal microbiota in only three human subjects (DOI:10.1371/journal.pbio.0060280), a sample size that some scientists outside of the field of microbial ecology would consider extremely too low to make any useful inferences.

RFI and its abbreviation: Done.

The sentence “The results suggest that the abundance of some ruminal microbes is associated with feed efficiency” is accurate and we did not find a better way to say that the results serve as a cautionary tale for the interpretation of results obtained with a single DNA extraction method.

INTRODUCTION

The sentence “more details are still needed to better understand the nature of this relationship” was changed to “more details are still needed to better understand the nature and characteristics of this relationship”, which serves to connect with the next paragraph.

Your suggestion was used to complement the sentence “additional factors” with anatomical or physiological.

Based on your comment and the comments from other reviewers, we decided to add the full list of ingredients as Supplementary Information.

We added a brief description of the GrowSafe system.

We added the QIIME plugin used for diversity analyses.

Very good question. We do not know the relevance of these taxa in gut microbial ecology. For instance, Ley et al (2005) suggested that gut Cyanobacteria may represent descendants of nonphotosynthetic ancestral cyanobacteria that have adapted to life in animal gastrointestinal tracts (DOI:10.1073 pnas.0504978102). We added a sentence about this.

About the analysis of volatile fatty acids, the steel collector tip in the SELEKT system (now updated in material and methods) does not allow the passage of solids and one centrifugation step is therefore enough to allow the use and collection of liquid for this analysis. The reference is a manual on Laboratory Procedures in Animal Nutrition Research designed by M.L. Galyean from Texas Tech University.

The version (release) of SAS University Edition is now included.

The sentence about independent variables was clarified.

RESULTS

The animal with the highest RFI value (4.76) was not selected because that value was further than 3 standard deviations from the mean, as declared in the manuscript.

We think that mean (minimum, maximum) is better to reflect the values provided, and this was declared in the bottom of the table.

Those two words are mentioned only once in the entire manuscript and to include the abbreviations we would nonetheless have to also include the full name.

The first sentence of the subsection 3.3 was rephrased to make it more understandable.

The first sentence of subsection 3.4 was rephrased to make it more understandable.

The comment about Anaerolineae was moved to discussion.

In our experience, the comment about the differences within phylogenetic levels may not be necessarily true, because the different members of each phylum (or class or any other phylogenetic level) often display different behaviors.

DISCUSSION

The entire first paragraph of discussion was removed.

We agree with the reviewer in that the word “phenomenon” is not accurate, and therefore we changed it to “the effect of DNA extraction is well known in gut microbial ecology”.

We agree with the reviewer in that the phrase “to meditate” is not accurate, and therefore we rephrased the sentence.

Round 2

Reviewer 1 Report

  • The author has revised the article according to the comments.

Reviewer 3 Report

The authors responded adequately to the raised points. The manuscript has been significantly improved and now permits publication in Microorganisms.

Reviewer 4 Report

Dear authors, it seems to me that this manuscript has great relevance in the scientific world. I think the authors made the suggested changes; therefore, I recommend approval.

Reviewer 5 Report

work has been significantly improved.